# High-Performance Ultraviolet Photodetector Based on a Zinc Oxide Nanoparticle@Single-Walled Carbon Nanotube Heterojunction Hybrid Film

**DOI:** 10.3390/nano10020395

**Published:** 2020-02-24

**Authors:** Myung-Soo Choi, Taehyun Park, Woo-Jae Kim, Jaehyun Hur

**Affiliations:** 1Department of Chemical and Biological Engineering, Gachon University, Seongnam-si, Gyeonggi-do 13120, Korea; choimysoo@gmail.com (M.-S.C.); chemtry0430@naver.com (T.P.); 2Department of Chemical Engineering and Material Science, Ewha Womans University, Seoul 03760, Korea

**Keywords:** zinc oxide nanoparticle, carbon nanotube, ultraviolet photodetector, hybrid film, heterojunction

## Abstract

A hybrid film consisting of zinc oxide nanoparticles (ZnO NPs) and carbon nanotubes (CNTs) is formed on a glass substrate using a simple and swift spin coating process for the use in ultraviolet photodetectors (UV PDs). The incorporation of various types of CNTs into ZnO NPs (ZnO@CNT) enhances the performance of UV PDs with respect to sensitivity, photoresponse, and long-term operation stability when compared with pristine ZnO NP films. In particular, the introduction of single-walled CNTs (SWNTs) exhibits a superior performance when compared with the multiwalled CNTs (MWNTs) because SWNTs can not only facilitate the stability of free electrons generated by the O_2_ desorption on ZnO under UV irradiation owing to the built-in potential between ZnO and SWNT heterojunctions, but also allow facile and efficient transport pathways for electrons through SWNTs with high aspect ratio and low defect density. Furthermore, among the various SWNTs (arc-discharged (A-SWNT), Hipco (H-SWNT), and CoMoCat (C-SWNT) SWNTs), we demonstrate the ZnO@A-SWNT hybrid film exhibits the best performance because of higher conductivity and aspect ratio in A-SWNTs when compared with those of other types of SWNTs. At the optimized conditions for the ZnO@A-SWNT film (ratio of A-SWNTs and ZnO NPs and electrode distance), ZnO@A-SWNT displays a sensitivity of 4.9 × 10^3^ % with an on/off current ratio of ~10^4^ at the bias of 2 V under the UV wavelength of 365 nm (0.47 mW/cm^2^). In addition, the stability in long-term operation and photoresponse time are significantly improved by the introduction of A-SWNTs into the ZnO NP film when compared with the bare ZnO NPs film.

## 1. Introduction

Ultraviolet (UV) photodetectors (PDs) have drawn much attention because of their wide range of applications such as UV monitoring in the environment, flame sensing, missile launch detection, biological sensing, and space communications [1,2,3]. Although silicon photodiodes have been commonly used as excellent UV detection devices, they generally exhibit some inherent limitations: (i) they require appropriate filters to block the photons in the visible and infrared ranges, (ii) they need to be cooled down to avoid changes in the dark current, and (iii) they are susceptible to degradation under exposure to UV light with energies that are much higher than those of the semiconductor bandgap [4,5,6,7,8].

ZnO-based PDs are promising candidates as alternatives for silicon-based PDs because *n*-type ZnO semiconductors allow an intrinsic visible-blindness (associated with a direct large band gap of 3.3 eV) and room-temperature operation. In addition, they have the advantages of simple synthetic methods, excellent chemical stability, and environmental friendliness [9,10,11,12,13,14]. Consequently, many researchers have developed ZnO-based ultraviolet photodetectors (UV PDs) with different ZnO nanostructures, such as nanoparticles (NPs) [15,16], nanorods [17], and nanowires [18,19], through various synthetic processes. Most of early studies on ZnO-based PDs have mainly focused on controlling various ZnO nanostructures to increase the surface-to-volume ratio that allows rich oxygen molecules adsorbed on to the ZnO surface and higher carrier concentrations that eventually enhance the sensitivity of UV PDs [20,21,22,23,24,25,26]. The photocurrent generation mechanism is regulated by this O_2_-mediated free-electron generation when ZnO is used standalone as an active material in UV PD devices regardless of the morphology of and synthetic process for ZnO. In this mechanism, the electronegative O_2_ molecules adsorbed on ZnO surface can be desorbed upon UV light because of the charge neutralization by the generated holes, resulting in the free electrons on ZnO. During this process, free-electrons can flow under external potential bias, which is revealed as photocurrent. However, these types of devices generally suffer from a poor sensitivity (low on–off ratio) and photoresponse behavior (a long rise and decay time) that is associated with the instability of free electrons which can recombine with holes that have not been completely paired with electronegative O_2_ and slow O_2_ adsorption/desorption processes, respectively [27].

In the recent years, the introduction of a p-type semiconductor to *n*-type ZnO has been proposed to overcome this problem, because the formation of a *p-n* heterojunction can avoid the slow O_2_-mediated photocurrent generation and directly use both electron and hole as photocarriers. Along this line, Cai et al. reported that the ZnO/poly(*N*-vinylcarbazole) bilayer film prepared using sequential spin coating exhibited a faster photoresponse to UV light than pristine ZnO [28]. A similar concept was demonstrated by Mridha et al. in which they employed polyaniline as a p-type organic semiconductor to active ZnO to form a heterojunction that exhibited an enhanced the photoresponse behavior than in the case of bare ZnO [29]. In addition to the organic *p*-type materials, the introduction of inorganic graphdiyne and perovskite CsPbBr3 has also been demonstrated to improve the performance of ZnO-based UV PDs [30]. On the other hand, some other interesting effects like scattering phenomena upon decorating diatom frustules on ZnO NPs [31,32].

Carbon nanotubes (CNTs) are 1-dimensional carbon allotropes that have a high aspect ratio and electric conductivity [33]. CNTs can be categorized into single-walled carbon nanotubes (SWNTs) and multiwalled carbon nanotubes (MWNTs) depending on the number of graphene cylinders: a single nanotube for SWNTs and several concentric nanotubes for MWNTs. In general, SWNTs possess the combined behavior of semiconductors and metals, whereas MWNTs exhibit a sorely metallic behavior because they are zero-gap metals. Furthermore, the SWNTs can also be divided further depending on their chirality and synthetic methods. Largely, there are three types of SWNTs based on the synthetic ways: arc-discharged SWNTs (A-SWNTs), high-pressure carbon monoxide SWNTs (HipCO SWNTs or H-SWNTs), and cobalt–molybdenum catalyst SWNTs (CoMoCat SWNTs or C-SWNTs). These SWNTs have different mechanical, electrical, and optical properties [34,35,36,37,38,39]. Therefore, the introduction of different types of CNTs to ZnO provides an excellent opportunity to explore systematic studies on the enhancement of ZnO-based UV PDs. In this context, Li et al. demonstrated the *p*-SWNT/*n*-ZnO vertical heterostructures for the high-performance UV PDs, but they did not investigate the effect of different types of SWNTs on the ZnO-based UV PDs [39]. In addition, although the combination of SWNT and ZnO has been also demonstrated by Ates et al. for the application of UV PDs, the SWNTs in this work were not utilized in the active layer but as transparent electrodes for ZnO nanowires [19]. Despite the previous approaches using both SWNTs and ZnO, none of these works has focused on the systematic study of the effect of different types of SWNTs introduction on the performance of ZnO-based UV PDs. Therefore, the fabrication of ZnO@CNT hybrid films with different types of CNTs will provide the good opportunity to study the role of different CNTs on the ZnO-based UV PDs which can be used to enhance the performance of UV PDs. As aforementioned, semiconducting *p*-type SWNTs are expected to create a built-in electric field near the interfaces (that is, the formation of *p-n* heterojunctions) with *n*-type ZnO which may stabilize the free electrons and offer effective transport pathways for the photocarriers simultaneously. In contrast, MWNTs cannot contribute to the stabilization of free-electrons on ZnO because of the metallic property, although they can also function as a guide for the photocarrier movement.

Herein, we report the preparation of a high-performance ZnO NP@CNT hybrid film for UV PDs using a simple spin coating process. The photoelectrical properties of the ZnO NP@CNT hybrid film are systematically investigated with different types of CNTs (MWNTs, A-SWNTs, H-SWNTs, and C-SWNTs) and their concentration in the composite. We demonstrate that ZnO@A-SWNT exhibits the best performance in terms of sensitivity, photoresponses (the fastest rise and decay times), and reproducibility in long-term operation when compared with those of other ZnO@CNT hybrid films and pristine ZnO NP films. The sensitivities (on–off ratio) of ZnO@A-SWNT (ratio of 10:1) are 950 and 310 for the UV wavelengths of 254 nm and 365 nm, respectively, at the light intensity of 1.25 mW/cm^2^. The respective rise, decay, and recovery times are 11.3, 6.0, and 2.7 s, which are much faster than those of the pristine ZnO NPs film (14.4, 33.6, and 11.8 s). The enhanced performance by ZnO@A-SWNT is attributed to the combination of improved stability of free electrons by the *p-n* heterojunction formation (hole depletion and enriched electrons between ZnO and semiconducting SWNT) and improved electric transport through a highly conductive A-SWNT matrix with low defect density, which effectively guides the photocarrier movement.

## 2. Materials and Methods 

### 2.1. Materials 

Zinc acetate dihydrate (Zn(CH_3_COO)_2_·2H_2_O, product #:1001738977: ZA), methanol (99.9 wt.%), butanol (99.8 wt.%), and C-SWNT (SG56, product #:704148) were purchased from Sigma Aldrich (St. Louis, MO, USA). A-SWNT (AST-100F, 5–10 µm in length, 1.3–1.5 nm diameter) was provided by Hanwha Nanotech (Daejeon, Korea). H-SWNT (HR27-122, 0.1–1 μm in length, 0.8–1.0 nm diameter) was bought from Nanointegris (Quebec, QC, Canada). MWNTs (30–50 μm in length, 15–40 nm in diameter) were supplied by Hanwha Nanotech. Co. Ltd. (Daejeon, Korea). Potassium hydroxide (KOH, 95%, product #: P1446) was produced by Samchun (Seoul, Korea). A glass substrate (15 mm × 15 mm) was manufactured from RnD Korea (Seoul, Korea).

### 2.2. Synthesis of ZnO NPs

ZnO NPs were synthesized by the condensation and hydrolysis of ZA using KOH in methanol. For this experiment, two different solutions were separately prepared: the first solution with 13.4 mmol ZA dissolved in 125 ml methanol at 60 °C, and the second solution with 23 mmol KOH dissolved in 65 mL methanol at 25 °C, respectively. Then, the KOH solution was injected into the ZA solution at 60 °C at a constant volumetric flow rate (6.5 mL/min) with vigorous stirring. After 2 h of reaction, the solution was changed from being transparent to whitish and turbid, indicating the formation of ZnO NPs in the solution (pH was 8–9). After completing the reaction, ZnO NPs were washed by repeating the centrifugation process (4000 rpm for 20 min), decanting the supernatant, and dispersing in methanol at least 3 times. ZnO NPs obtained was ~500 mg. As the final step, ZnO NPs were dispersed in the co-solvent of butanol and methanol (1:1, v/v) at a concentration of 10 mg/ml. The pH of this suspension was measured to be 6, indicating that the final ZnO NP suspension was at nearly neutral pH [40].

### 2.3. Preparation of a UV Sensor Based on ZnO NPs/CNT Hybrid Films

ZnO NP/CNT solutions were prepared by dispersing four different types of CNTs in the ZnO NP suspension: (i) A-SWNT, (ii) H-SWNT, (iii) C-SWNT, and (iv) MWNTs. Table 1 provides the detailed information for individual types of CNTs. To investigate the effect of the ZnO/CNT ratio, the concentration of CNTs added to the ZnO suspension (2 mL at a concentration of 10 mg/mL) was varied as 0.5, 0.7, 1.0, 2.0, and 5.0 mg (resulting in the weight ratios of 40:1, 28:1, 20:1, 10:1, and 4:1 between ZnO and CNTs). The dispersion of ZnO/CNT solution was performed with a tip sonicator (130 W ultrasonic processors, Ningbo SCIENTZ Biotechnology Corp., Zhejiang, China) for 5 min to prevent aggregation. To prepare the ZnO NP/CNT hybrid film, a glass substrate was washed with acetone, isopropyl alcohol, and DI water, respectively, followed by drying using a N_2_ blowing gun. Subsequently, the substrate was cleaned with UV-ozone (UV SMT, 25 W, UV SMT corp., Gyeonggi, Korea) for 30 min. Then the ZnO NP/CNT solution was spin-coated at 2500 rpm for 40 s, followed by annealing in the air at 120 °C for 20 min. The spin coating and annealing process was repeated up to 5 times to obtain the optimum thickness for the ZnO@CNT film. Two Ag electrodes (1 mm × 3 mm, thickness of 150 nm) were deposited using a thermal evaporator (DKOL04-02-T, MBE Komponenten Corp., Weil der Stadt, Germany). The distance between the electrodes is adjusted from 1 mm to 9 mm using shadow masks with different configurations. As a control sample, an UV sensor with the pure ZnO NPs film was separately fabricated using the same method.

### 2.4. Characterization and Device Measurement

The structure of the as-synthesized ZnO was examined using X-ray diffraction (XRD; D/MAX-2200 Rigaku, Tokyo, Japan). The optical absorbances of ZnO and ZnO/CNT were obtained using a UV–Vis spectrophotometer (LAMDA 750, Perkin Elmer, Waltham, MA, USA). The morphologies and elemental distribution of the as-prepared samples were observed using high-resolution transmission electron microscopy (HRTEM, Tecnai G2F30, Thermo Fisher Scientific, Waltham, MA, USA) coupled with energy-dispersive X-ray (EDX) spectrometry. The film thickness was analyzed using field emission scanning electron microscopy (FE-SEM, S-4700, Mono CL, Hitachi, Tokyo, Japan). To measure the characteristics of UV sensors, the current–voltage (I–V) and current–time (I–t) were measured using a Keithley 2400 source meter in the dark and under UV illumination. The UV light source used was a UV lamp (UV-5NF, MiniMAX, Spectroline, Westbury, NY, USA) with wavelengths of either 254 or 365 nm. The UV light intensity was adjusted by controlling the distance between the light source and the sample, and the power density was measured using a spectrometer (CAS140CT, Instrument Systems, Munich, Germany). In the UV-sensing measurement, only the active area (1 mm × 3 mm between the two electrodes) was illuminated by masking the remaining area. For the repeated on/off operations of UV light, a mechanical shutter was used with UV light switched on.

## 3. Results and Discussion

### 3.1. Characterization of the As-Prepared ZnO NPs and the ZnO NP/CNT Film

Figure 1a presents the powder XRD patterns of the synthesized ZnO NPs. The typical peaks observed at 31.8°, 34.5°, 36.3°, 47.5°, 56.6°, and 62.9° correspond to the (100), (002), (101), (102), (110), and (103) planes of crystalline ZnO (PDF# 27-1402) [41,42]. No other distinctive peaks were observed, confirming that no impurities were produced. The UV–Vis absorbance spectra of ZnO NPs, ZnO@MWNT films, and ZnO@SWNT films are displayed in the wavelength range of 250 to 550 nm (Figure 1b). All these samples show selective absorbance in the UV range (250–360 nm), which justifies the applications to the UV PDs. Despite the absorbance of visible and infrared range from pure SWNTs (inset in Figure 1b), the visible blindness is well-maintained because of the small amount of SWNTs relative to the ZnO in the ZnO/SWNT hybrid film [43].

The microscopic powder morphology and elemental distribution of the as-prepared ZnO NPs and ZnO@SWNT were observed by HRTEM imaging (Figure 2). As shown in Figure 2a,b, as-prepared ZnO NPs exhibited several nanocrystallites of average size of 5–10 nm and an interplanar spacing of 0.25 nm corresponding to the (101) plane of ZnO (the selective area diffraction pattern is shown in the inset in Figure 2b). For ZnO@SWNT, the presence of SWNTs effectively surrounding the ZnO NPs without much aggregations was also confirmed (Figure 2c,d). This morphology is advantageous for UV PDs because the ZnO@SWNT hybrid film can increase the surface area of the heterojunction and effectively enhances the electron transport under the UV exposure. Figure 2e,f confirmed the homogeneous distribution of constituent elements (Zn and O for ZnO NPs, and Zn, O, and C for ZnO@SWNT, respectively).

### 3.2. Characterization of ZnO and ZnO@CNT Film

Figure 3 shows the SEM images of the bare ZnO film (Figure 3a,b, 7 rounds of spin coating, thickness of 165 nm) and ZnO@CNT films (Figure 2c–f) by varying the number of spin coating (1, 3, and 5). The spin coating produced a well-defined film without much variation in the thickness for the bare ZnO film (Figure 3a). The formation of quasi-close-packed arrangement of ZnO NPs (Figure 3b) via spin coating is associated with the uniform size of ZnO NPs (5–10 nm as shown by the TEM analyses in Figure 2) and a suitable solvent (co-solvent of butanol and methanol) of appropriate viscosity. Even though CNTs were added to ZnO NPs, a high-quality ZnO@CNT film was still formed (Figure 2c–f). The thickness of the film could be systematically controlled by the number of spin coating; the thicknesses were 25, 67, and 165 nm for 1, 3, and 5 times of spin coating, respectively. Although the photocurrent can be increased by increasing the number of spin coating, the variation in thickness in a given spin coating condition became more pronounced for excessively thick films (beyond 5 rounds of spin coating), which made it difficult to obtain reproducible photocurrent measurements. Therefore, the optimal thickness of the ZnO@CNT film was controlled to be ~165 nm with 5 rounds of spin coating.

Figure 4a presents the schematic illustration of the fabrication of the UV sensor device. On the glass substrate, a well-defined ZnO@CNT film was prepared by repetitive spin coating. Thermal annealing (120 °C at 20 min) was necessary to enhance the crystallinity of ZnO NPs and to stabilize the interfaces between ZnO NPs films. Two Ag electrode terminals were deposited using a thermal evaporator with a shadow mask that defined the active area. As shown in Figure 4b, the UV sensor device appeared generally transparent because of the transparency of the substrate (glass) and the active region (ZnO@CNT).

### 3.3. Electrical Characteristics of ZnO, ZnO@MWNT Film, and ZnO@SWNT Film

Figure 5a,b displays the I–V characteristics measured in the dark and under UV light illumination (254 and 365 nm) at a light intensity of 1.25 mW/cm^2^ over a voltage bias of 2 V. The thicknesses of the film were controlled to be all ~165 nm for all samples. The linear I–V relationship with forward and reverse biases indicates the ohmic contact between ZnO@CNT and Ag electrodes. The ohmic contact behavior allows a negligible contact resistance at the metal–semiconductor interface and the linear relationship between bias and photocurrent. The sensitivity (S) of the device to the UV light can be calculated using the following equation [44],
(1)$$S=\frac{I_{\mathrm{uv}}-I_{d}}{I_{d}}$$
where I_uv_ and I_d_ denote the current under UV illumination and in the dark, respectively. The S value was the highest for ZnO@SWNT, followed by ZnO@MWNT and ZnO; they were 950, 66, and 50 for UV light of 254 nm (1.25 mW/cm^2^) and 310, 24, and 22 for 365 nm (1.25 mW/cm^2^), respectively. These trends were consistent regardless of the light intensity.

The enhanced sensitivity in the presence of SWNTs can be explained by the different photocurrent mechanisms for MWNTs and SWNTs in the ZnO-based UV sensor. As shown in Figure 6, the photocurrent for ZnO@MWNT is mainly governed by the oxygen desorption mechanism under the UV irradiation, which induces the hole depletion, thus generating free electrons (Figure 6a). The free electrons generated in this process can move between two electrodes under the applied bias [45,46,47]. In this mechanism, the presence of MWNTs, which are metallic, simply provides pathways for the free electrons to move from ZnO to Ag electrode, thus facilitating electron transport along the MWNTs. However, there are still some holes which did not participate in pairing with negatively-charged oxygen remained in ZnO NPs. Possibly, these remained holes can recombine with electrons, resulting in the decrease in photocurrent. On the other hand, in the case of ZnO@SWNT, of which SWNTs have both metallic and semiconducting characters with band gaps, the holes generated under the UV light are partially trapped with charged oxygen, and free electrons are extracted mainly through the metallic SWNTs (~33%) under the applied voltage bias between two electrodes as in the case of ZnO@MWNT (Figure 6b). At the same time, the remained holes which did not pair with charged oxygen can be additionally extracted toward adjacent p-type semiconducting SWNT (~66%), of which HOMO level exists above valence band of ZnO [48,49,50,51]. This additional hole extraction prevents the recombination between holes and electrons, which is considered as a main factor that significantly enhances the sensitivity of ZnO@SWNT over that of ZnO@MWNT. From our separate measurements for another control sample (bare SWNTs or MWNTs without ZnO), there was no photoresponse for UV light (Figure 5c,d). This was because CNTs cannot produce photocurrent by itself because free electron cannot be generated even if CNTs themselves absorb the light in the UV region [52]. Although the mechanism of free electrons based on oxygen desorption under the UV light has been dominantly considered for most of the ZnO-based UV sensors with a lateral geometry (in-plane) of electrode terminals [53], the performance improvement in our hybrid ZnO@SWNT film can be attributed to the combination of two different mechanisms (i.e., electron transport through metallic SWNT and hole extraction toward semiconducting SWNT as shown in Figure 6b).

Another factor that contributes to the sensitivity is the structural quality of the CNT matrix. Figure 7 compares the Raman spectra between SWNTs and MWNTs. Both SWNTs and MWNTs show two peaks at ~1350 cm^−1^ (D-band) and 1580 cm^−1^ (G-band), respectively. The G-band and D-band are characterized by the sp_2_ hybrid carbon–carbon bonds of graphene and their defects or disorder, respectively. The I_D_/I_G_ ratio reflects the number of sp_3_ carbon atoms present at the defect site and the open end [54]. The MWNTs, consisting of composite graphite layers, showed a strong intensity in the D band due to the large number of defects sites (I_D_/I_G_ = 1.16). However, the low I_D_/I_G_ ratio of SWNTs (I_D_/I_G_ = 0.20) indicates a low level of structural defects. In general, graphene with low crystalline quality has low electrical conductivity or poor mechanical strength because of the defects or oxygen sites [55,56]. Therefore, it can be expected that the electrons produced at ZnO NPs under the UV irradiation can be more sustainable and can efficiently move along SWNTs without much loss than MWNTs between Ag electrodes. In addition to the effect of structural quality of CNT, for the same total weight of CNT, the SWNT network will be much denser and in closer vicinity to each ZnO NP as one MWNT is equal by weight to hundreds of SWNTs. This is considered to be another factor that is beneficial to enhance the sensitivity.

Figure 8 compares the I–V relationship for ZnO@SWNT with three different types of SWNTs (A-SWNTs, H-SWNTs, and C-SWNTs) along with their energy band diagrams, denoted as ZnO@A-SWNT, ZnO@H-SWNT, and ZnO@C-SWNT (ZnO/SWNT = 20:1, w/w), respectively, under the 254 and 365 nm UV light illumination (0.47 mW/cm^2^). In general, SWNTs can be classified depending on the synthesis methods. A-SWNTs, H-SWNTs, and C-SWNTs are three representative SWNTs that are synthesized by the arc-discharge, high-pressure carbon monoxide disproportionation, and chemical vapor deposition using catalysts such as Co and Mo. These different SWNTs possess typical physical dimensions and chiralities that affect their mechanical, electrical, and optical properties. The characteristics of these different SWNTs are summarized in Table 1. From the I–V measurements, the sensitivity was found to be the highest for ZnO@A-SWNT, followed by ZnO@H-SWNT and ZnO@C-SWNT for both 254 and 365 nm UV irradiation; the S values were 340, 60, and 8 for 254 nm UV light of (0.47 mW/cm^2^) and 100, 15, and 1.3 for 365 nm UV light (0.47 mW/cm^2^) at a bias of 2 V, respectively (Figure. 7a–c). These trends were consistent with those for the other SWNT concentrations (Appendix A). Because the photocurrent mechanism is governed by both (i) the free electron generation from the oxygen desorption on the ZnO surface followed by the electron transport through metallic SWNT and (ii) the additional hole extraction toward semiconducting SWNT due to the *p-n* heterojunction for all ZnO@SWNT samples, the sensitivity is associated with the combination of these two mechanisms simultaneously. Upon comparing the energy band for different SWNTs [48,49], ZnO@C-SWNT should be the most favorable in extracting holes at heterojunction under the UV light followed by ZnO@H-SWNT and ZnO@A-SWNT (Figure 7d–f). However, the measured sensitivity was opposite to this trend. This suggests the sensitivity is highly correlated with other factors among SWNTs. Because the defect densities in different SWNTs are similar, as revealed by Raman spectroscopy (Appendix A), the electric conductivity of metallic SWNT in each SWNT is the major factor that determines the sensitivity. Indeed, as shown in Table 1, the sheet resistance was the lowest for A-SWNTs (the thicknesses for all SWNT film were controlled to be 100 nm), suggesting that the electron transport will be most efficient for ZnO@A-SWNT. The effect of band structure for different SWNTs seems to be relatively small, because of their trivial bandgap differences compared with that of ZnO (Figure 8d–f). Another important factor that can contribute to the sensitivity of SWNTs is the aspect ratio of SWNTs. The aspect ratios of SWNTs were 3300–7700, 100–1250, and 120–1430 for A-SWNTs, H-SWNTs, and C-SWNTs, respectively. The highest aspect ratio for A-SWNT leads to a reduced percolation threshold and more efficient carrier transport pathways for the given SWNT concentration.

Figure 9 displays the I–V characteristics under the effect of the concentration of A-SWNTs in ZnO@A-SWNT (Figure 9a) and electrode distance (Figure 9b). The sensitivity generally increased with the increase in the concentration of A-SWNTs (Figure 9a). This is because of the increased number of electron–hole pairs at the *p-n* heterojunction and more A-SWNT pathways for the photogenerated electrons with the increase in A-SWNTs. However, the excessive A-SWNTs mixed with ZnO NPs adversely affected the sensitivity because of the poor dispersion of A-SWNTs in the ZnO/A-SWNT solution, which induced the aggregated domains on the film after spin coating (Figure 9a). This behavior was commonly observed regardless of the types of SWNTs (Appendix A (solution) and Appendix A (film)). The sensitivity was maximized only at the optimum mass ratio between ZnO NPs and SWNTs (Figure 9a). The reduction in electrode gap increased the sensitivity because of the reduced pathway on which the photocarriers travel under the 254 nm UV irradiation (Figure 9b).

Finally, photoresponse for the ZnO@A-SWNT UV sensor was measured to investigate the photocarrier interaction between ZnO NPs and A-SWNTs. Figure 10a shows the spectral photoresponsivity (left y-axis) and detectivity (right y-axis) as a function of light wavelength for the ZnO@A-SWNT UV PD under the bias of 10 V. The responsivity (R) and detectivity can be calculated based on the following equation,
(2)$$R\left(\lambda \right)=\frac{I_{\mathrm{uv}}\left(\lambda \right)-I_{d}\left(\lambda \right)}{P\left(\lambda \right)\times A}$$
(3)$$D=\frac{R\left(\lambda \right)}{{\left(2qI_{d}\left(\lambda \right)\right)}^{1/2}}$$
where R(λ), I_uv_(λ), I_d_(λ), P(λ), A, D, and q denote the responsivity (A/W), the current under UV illumination at the wavelength of λ, the current in the dark (A) at λ, the light intensity (W/cm^2^), the active area (cm^2^), the detectivity (cm Hz^1/2^ W^−1^ or Jones), and the elementary charge of a single electron, respectively. The maximum responsivity and detectivity were measured to be ~0.04 A/W and 7.1 × 1012 cm Hz^1/2^ W^−1^ at 250 nm, followed by a gradual decrease in responsivity and detectivity with the increase in the light wavelength, showing a similar trend to that observed in the absorbance spectrum (Figure 1b). The cut-off edge wavelength was measured to be ~380 nm for the ZnO@A-SWNT UV PD, indicating the visible blindness of the device [57,58,59,60]. Figure 10b,c shows the photocurrent as a function of time for bare ZnO and ZnO@A-SWNT under 365 nm UV irradiation with 0.47 mW/cm^2^ at a bias of 2 V. Here, the rise time refers to the time to reach 90% of the maximum photocurrent after UV light is switched on and the decay time corresponds to the time to reach 10% of the maximum photocurrent after UV light is switched off [61,62]. The rise times of bare ZnO and ZnO@A-SWNT devices were 14.4 s and 33.6 s, respectively, and the decay times were 11.3 s and 6.0 s, respectively, resulting in a much faster photoresponse of ZnO@A-SWNT because of the existence of highly conductive A-SWNTs surrounding Zn NPs. The recovery time (defined as the time for photocurrent to decrease to 37% of the maximum current state) was ~11.8 s for ZnO NPs, but ~2.7 s for ZnO@A-SWNT. The reduction in rise time, decay time, and recovery time for ZnO@A-SWNT relative to those of bare ZnO NPs is attributed to the enhanced transport of photoinduced electrons in the ZnO@A-SWNT hybrid structure. Figure 10d,e compares the photoresponses of ZnO NPs and the ZnO@A-SWNT UV sensor for a repeated on/off operation under the 365 nm UV light (0.47 mW/cm^2^) at 2 V bias. ZnO@A-SWNT displayed a stable on/off cyclic performance with enhanced sensitivity, increased signal-to-noise ratio, and better reproducibility relative to those of the bare ZnO. Specifically, at the given measurement conditions, the sensitivity of ZnO@A-SWNT was higher than that of ZnO by a factor of ~45. The on/off current ratio for ZnO@A-SWNT remained at ~80% after 20 cycles, which is much higher than that of ZnO (~63%), indicating a much more reliable on/off cyclic performance for ZnO@A-SWNT. This measured performance of our ZnO@A-SWNT is outstanding in comparison with those of other ZnO-based UV PDs reported in the literature (Table 2) [60,61,62,63,64]. Even comparing with other photoconductor-type UV sensors using different active materials, our ZnO-based device shows good performance in terms of sensitivity and response time (Table 3). All these enhanced performances due to the introduction of A-SWNTs to the ZnO film are based on the effective combination of photocurrent generations: the formation of effective pathways for the photogenerated electrons and the flow of electrons and holes generated by the *p-n* heterojunctions between ZnO and A-SWNTs.

## 4. Conclusions

We demonstrated a high-performance UV PD using the ZnO@A-SWNT hybrid film prepared via a facile and fast spin coating process. The effect of introducing different types of CNTs (A-SWNTs, H-SWNTs, C-SWNTs, and MWNTs) on the performance of UV PDs was systematically investigated. The addition of semiconducting SWNTs to ZnO exhibited a superior performance when compared with the addition of metallic MWNT based on the dual mechanisms of enhanced photocarrier (electron) stability (*p-n* heterojunction between SWNTs and ZnO) and more efficient electron transport along the SWNTs with a low defect density. Among the different SWNTs, A-SWNTs showed the most suitable combination with ZnO because of their higher electric conductivity and higher aspect ratio, despite the most unfavorable band structure. As a result, the ZnO@SWNT hybrid film exhibited an excellent performance as a UV PD with improved sensitivity and photoresponse and long-term operation compared to other cases. At the optimized conditions, the ZnO@SWNT-based UV PD showed a high sensitivity (490), high on/off current ratio (> 100,000), and fast photoresponses (rise time of 11.3 s and decay time of 6.0 s) at a bias of 2 V under the UV light (365 nm, intensity of 0.47 mW/cm^2^), which is much superior to the performance of pristine ZnO NP film. Overall, our current study provides the new insight regarding the rationale strategy to enhance the lateral-type UV PD devices.

## Figures and Tables

**Figure 1 nanomaterials-10-00395-f001:**
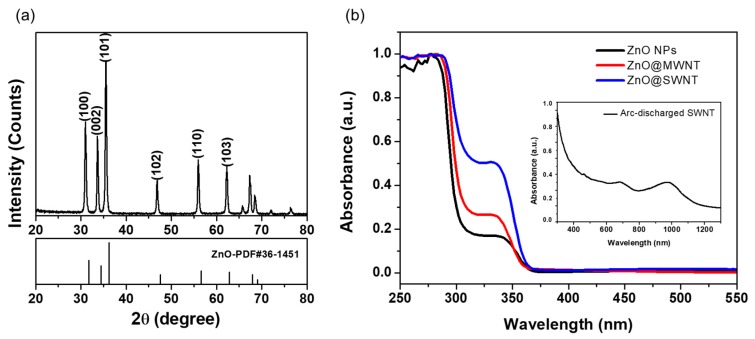
(**a**) XRD patterns of the ZnO NPs and (**b**) UV–Vis absorption spectra of ZnO NPs, ZnO@MWNT, and ZnO@SWNT. Inset shows the UV–Vis absorption spectrum of pure A-SWNT.

**Figure 2 nanomaterials-10-00395-f002:**
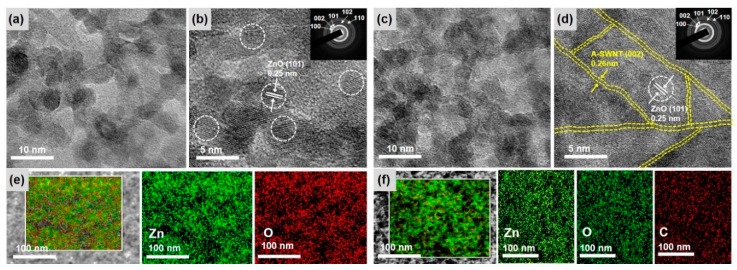
Bright-field TEM images of (**a**) ZnO NPs and (**c**) ZnO@SWNT; HRTEM images of (**b**) ZnO NPs and (**d**) ZnO@SWNT (SAED pattern are shown in the insets in Figure 2b,d); and STEM images with the corresponding EDS mapping images of (**e**) ZnO NPs structure and (**f**) ZnO@SWNT.

**Figure 3 nanomaterials-10-00395-f003:**
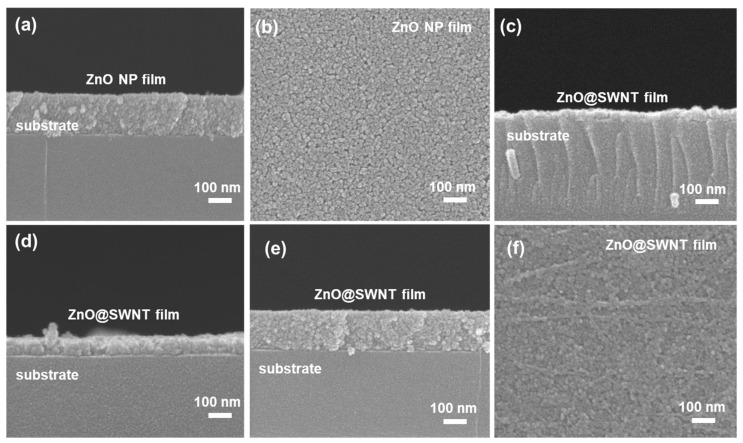
(**a**) Cross-sectional and (**b**) top-view SEM images of the ZnO NP film, (**c**–**e**) cross-sectional SEM images of ZnO@SWNT films spin-coated 1, 3, and 5 times, respectively, and (**f**) top-view of SEM image of the ZnO@SWNT film. The thickness of ZnO film (7 times of spin coating in Figure 3a) and ZnO@SWNT (5 times of spin coating in Figure 3e) were same as 165 nm.

**Figure 4 nanomaterials-10-00395-f004:**
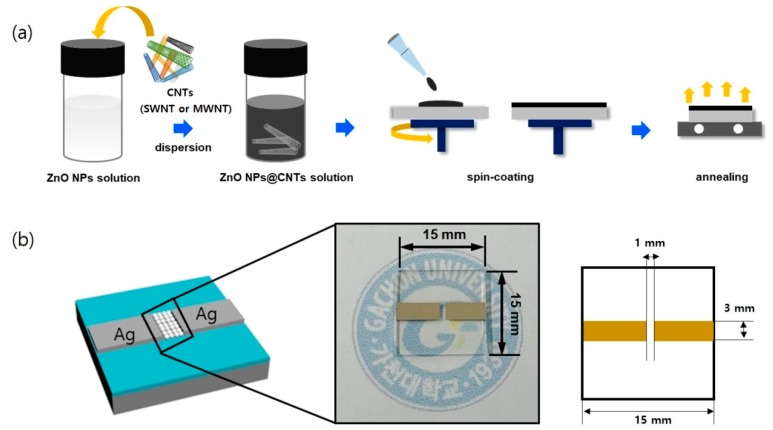
(**a**) Schematic of the ZnO@SWNT ultraviolet photodetector (UV PD) fabrication process. (**b**) Structure of the ZnO@SWNT UV PD device (the length and width of the glass substrate is 15 mm and 15 mm, respectively), with a portion shown in the paragraph.

**Figure 5 nanomaterials-10-00395-f005:**
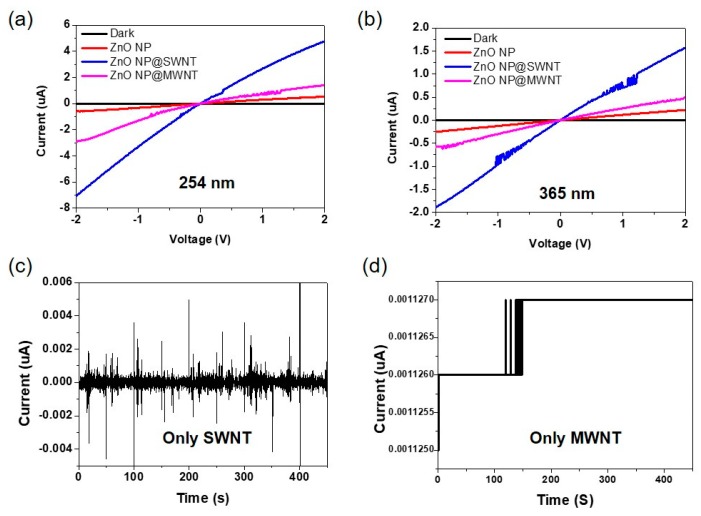
I–V curves of three different UV sensors (ZnO NPs, ZnO NP@SWNT, and ZnO NP@MWNT) under different wavelength UV irradiation (**a**) 254nm and (**b**) 365nm (1.25 mW/cm^2^), and photoresponse of UV sensors made of (**c**) only A-SWNT and (**d**) MWNT.

**Figure 6 nanomaterials-10-00395-f006:**
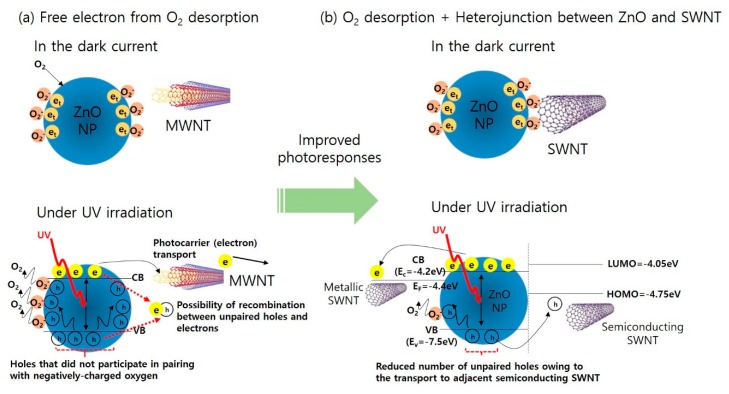
Illustration of the photocurrent mechanism for (**a**) ZnO@MWNT, free-electron generation from oxygen desorption under UV light, and (**b**) ZnO@SWNT, combination of oxygen desorption and hole depletion (electron enrichment) at the *p-n* heterojunction under UV light.

**Figure 7 nanomaterials-10-00395-f007:**
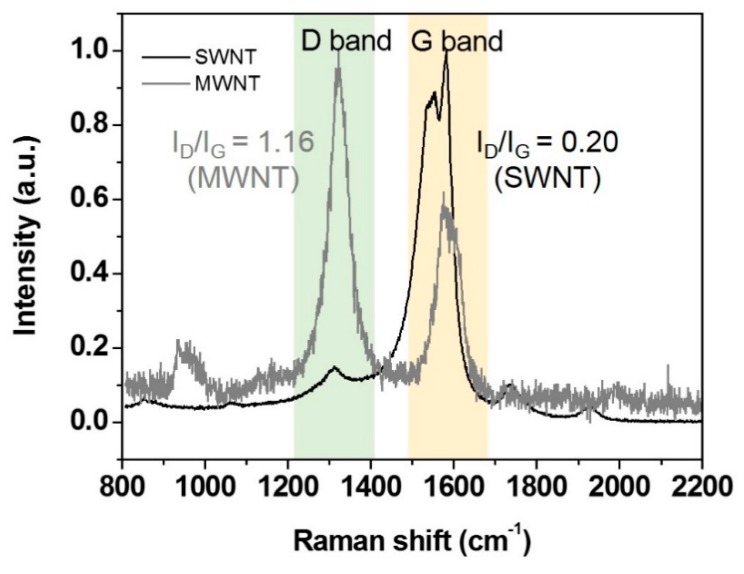
Comparison of Raman spectra between pristine SWNTs (black) and MWNTs (gray).

**Figure 8 nanomaterials-10-00395-f008:**
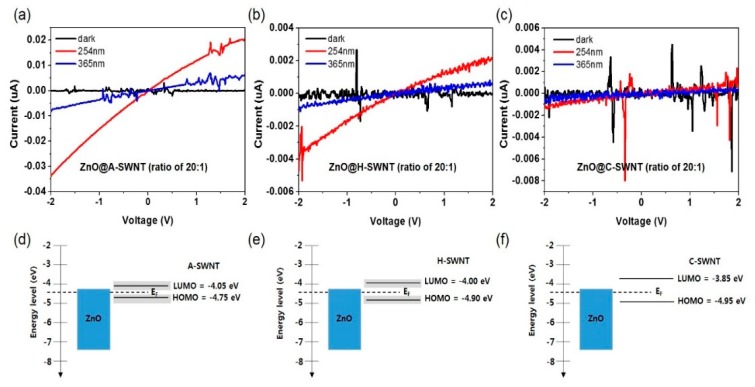
I–V curves for (**a**) ZnO@A-SWNT, (**b**) ZnO@H-SWNT, and (**c**) ZnO@C-SWNT(ZnO/SWNT = 20:1, w/w) under 254 and 365 nm UV light illumination (0.47 mW/cm^2^). Energy band diagram of (**d**) ZnO@A-SWNT, (**e**) ZnO@H-SWNT, and (**f**) ZnO@C-SWNT.

**Figure 9 nanomaterials-10-00395-f009:**
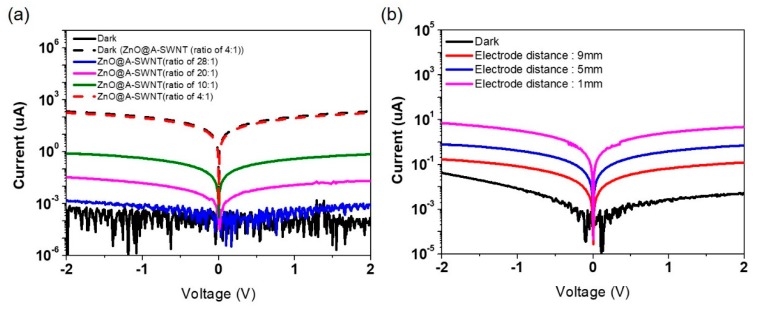
I–V characteristics of (**a**) ZnO@A-SWNT PDs with varying concentrations of A-SWNTs: 28:1, 20:1, 10:1, and 4:1 (ZnO:A-SWNT, *w*:*w*). (**b**) I–V characteristics of ZnO@A-SNWT PDs at different electrode distances: 1 nm, 5 nm, and 9 nm. The wavelength of the UV light was 254 nm (0.47 mW/cm^2^).

**Figure 10 nanomaterials-10-00395-f010:**
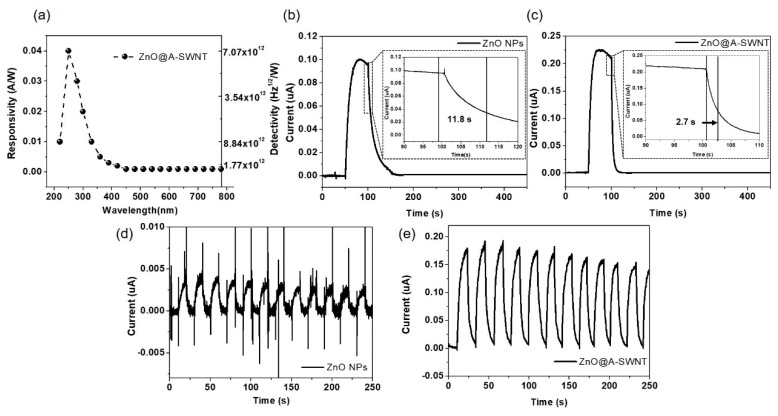
(**a**) Spectral responsivity of the ZnO@A-SWNT UV PD, photoresponses of the (**b**) ZnO NPs and (**c**) ZnO@A-SWNT PDs under UV irradiation (365 nm, 0.47 mW/cm^2^) at a voltage bias of 2V. The insets in (**b**) and (**c**) show the recovery times of ZnO NPs and ZnO@A-SWNT PDs, respectively. On/off switching properties with time-dependent measurements for (**d**) ZnO NPs and (**e**) ZnO@A-SWNT PDs.

**Table 1 nanomaterials-10-00395-t001:** Characteristics of various SWNTs: A-SWNTs, H-SWNTs, and C-SWNTS.

Materials	Length (μm)	Diameter (nm)	Aspect ratio	Property	Sheet Resistance (Ω/□)
A-SWNTs	5–10	1.3–1.5	3300–7700	~33% metallic (~67% semiconducting)	515
H-SWNTs	0.1–1	0.8–1.0	100–1250	~33% metallic (~67% semiconducting)	20620
C-SWNTs	0.1–1	0.7–0.8	120–1430	~10% metallic (~90% semiconducting)	34791

**Table 2 nanomaterials-10-00395-t002:** Comparison of performances for various ZnO-based UV PDs.

Material	Wavelength (nm)	Bias (V)	Intensity of Light (mW/cm^2^)	I_uv_-I_dark_	Rise Time (s)	Decay Time (s)	Ref.
ZnO nanowires	365	1	1.3	1000	40	300	[60]
ZnO microtubes	365	5	150	600	5.9	638	[62]
ZnO nanorods	365	5	-	85	>500	500	[63]
ZnO-coated ZnO arrays	365	3	150	1.25	120	180	[64]
ZnO@A-SWNT	365	2	1.25	49	11.3	5.97	This Work
	254	2	1.25	950	(@365nm)	(@365nm)	

**Table 3 nanomaterials-10-00395-t003:** Comparison of performances for various material-based UV PDs.

Material	Wavelength (nm)	Bias (V)	Intensity of Light (mW/cm^2^)	I_uv_-I_dark_	Rise Time (s)	Decay Time (s)	Ref.
SnO_2_ nanowire	250	3	-	10	>20	>10	[65]
ZnGa_2_O_4_ nanowire	350	5	0.53	130	15	>10	[66]
Graphene/TiO_2_	365	5	-	-	>150	>150	[67]
TiO_2_-ZnTiO_2_ nanowires	320	2	0.186	>300	>10	10	[68]

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
