# Peer review of "High-Performance Ultraviolet Photodetector Based on a Zinc Oxide Nanoparticle@Single-Walled Carbon Nanotube Heterojunction Hybrid Film"

_nanomaterials, 2020, doi:10.3390/nano10020395_

Round 1
Reviewer 1 Report
In this manuscript, the authors studied the UV photodetector using ZnO nanoparticles and carbon nanotubes. Reasonably high efficiency was experimentally obtained. Detail material characterization was conducted. And, electrical characterizations of photodetectors with different ZnO thickness and carbon nanotubes (single- and multi-walls) were systemically provided, which is very good. Hence, it is recommended for publication as it is.
Author Response
Point 1: In this manuscript, the authors studied the UV photodetector using ZnO nanoparticles and carbon nanotubes. Reasonably high efficiency was experimentally obtained. Detail material characterization was conducted. And, electrical characterizations of photodetectors with different ZnO thickness and carbon nanotubes (single- and multi-walls) were systemically provided, which is very good. Hence, it is recommended for publication as it is.
Response 1: We greatly appreciate reviewer 1 for the summary and positive comments.
Reviewer 2 Report
Manuscript reference number: nanomaterial 725262
Title: High-performance ultraviolet photodetector based on a zinc oxide nanoparticle@single-walled carbon nanotube heterojunction hybrid film
Myung-Soo Choi Jr et al
In this manuscript the authors studied how to enhance the performance of UV PDs with respect to sensitivity, photoresponse, and long-term operation stability with a hybrid film consisting of zinc oxide nanoparticles (ZnO NPs) and carbon nanotubes (CNTs). This paper is interesting and well organized but we suggest the authors to do major revisions of the manuscript so to correct the following parts:
Line 82: “….These SWNTs have different mechanical, electrical, and optical properties [32, 33]”
it is important here to complete such an overview quoting other fundamental related thermophysical properties of CNT as for example the emissivity and thermal properties. We suggest to quote some additional references as for example
1.G. Leahu, et al “Thermal Characterization of Carbon Nanotubes by Photothermal Techniques”, Int J Thermophys, 36, 1349 (2015).
2. Larciprete, M.C. International Journal of Thermal Sciences, 146, 106109, 2019
3. Macias, J.D., et al Applied Composite Materials, 26, 321, 2019.
In general the bibliography is rather incomplete and other interesting effects should be quoted as for example to distinguish the scattering phenomena from absorption phenomena in case of materials decorated with ZnO NP. These articles can be quoted for example:
Lamastra, F.R., et al Photoacoustic Spectroscopy Investigation of Zinc Oxide/Diatom Frustules Hybrid Powders, International Journal of Thermophysics, 39, 110, 2018,
Lamastra, F.R., et al Nanotechnology 28, Article number 375704 (2017)
Line 106: “…..are 9.5 Ń… 10^6 ….and 3.1 Ń… 10^4 %” . The percentage may confuse the reader. Instead of “10^4 %” is probably better to use “100 times”
Line 400: same comment as before “….showed a high sensitivity (4.9 Ń… 103), high on/off current ratio (> 104)”
Line 244 : The authors introduce in the text some figures calling them “Scheme 1”, “Scheme 1a”. We guess this is not the policy of the journal. We strongly recommend just to number it as a standard figure. In the caption for figure it is obviously mention that is a schematic representation.
In addition the use of the sublevel “i” and “ii” in the figure may create ambiguity. It is probably better much better to number the figures as (a), (b), (c), and (d)
Line 261 and 268, 306, 315. Supplementary figures S1, S2 and S3 are not in the main body of the article. Perhaps if these figures are important the authors can introduce and show them inside the main body with the standard numeration.
Fig.5, 7, and 8. The authors introduce a large number of figures 5,7,8, representing the total number of studied cases that can confuse the reader. We guess that the space can be more rational, by selecting only the figures showing the important effect, and by summarizing the other results by using tables and other tools.

Reviewer 3 Report
The paper systematically investigated with different types of CNTs (MWNTs, A-SWNTs, H-SWNTs, and C- SWNTs) and their concentration in the composite and demonstrated that ZnO@A-SWNT exhibits the best performance. The paper is highly interesting and worth publishing. The reviewer have a few minor comments.
- Some noises were observed in the measured results, For example, Fig. 5(d), 5(e), 5(f), 8 and 9(a). The authors should comment their source.
- The fabricated device seize should be added.
- Please comment the performance compared with other UV detectors not based on ZnO.
Round 2
Reviewer 2 Report
We guess that the manuscript has been improved after the revision, and could be ready to be published on Nanomaterials in the present form